# Plasma IL-1 and IL-6 Family Cytokines with Soluble Receptor Levels at Diagnosis in Head and Neck Squamous Cell Carcinoma: High Levels Predict Decreased Five-Year Disease-Specific and Overall Survival

**DOI:** 10.3390/cancers16081484

**Published:** 2024-04-12

**Authors:** Helene Hersvik Aarstad, Svein Erik Emblem Moe, Stein Lybak, Øystein Bruserud, Tor Henrik Anderson Tvedt, Hans Jørgen Aarstad

**Affiliations:** 1Department of Otolaryngology/Head and Neck Surgery, Haukeland University Hospital, 5009 Bergen, Norway; helene.hersvik.aarstad@haraldsplass.no (H.H.A.); svein.erik.emblem.moe@helse-bergen.no (S.E.E.M.); stein.lybak@helse-bergen.no (S.L.); 2Department of Surgery, Haraldsplass Deaconal Hospital, 5009 Bergen, Norway; 3Department of Clinical Medicine, Faculty of Medicine, University of Bergen, 5007 Bergen, Norway; 4Section for Haematology, Department of Medicine, Haukeland University Hospital, 5009 Bergen, Norway; 5Department of Haematology, Oslo University Hospital, Rikshospitalet, 0372 Oslo, Norway; totved@ous-hf.no

**Keywords:** head and neck squamous cell carcinoma, HPV, IL-6, IL-31, gp130, TNM stage

## Abstract

**Simple Summary:**

In cancer, including head and neck cancer (HNC), certain key players in the body’s inflammatory process, such as the interleukin (IL)-6 and IL-1 systems, are associated with outcomes. We studied some of these key players in a subgroup of HNC patients. From one blood sample measured at diagnosis, we determined high levels of molecules from members of the IL-6 and IL-1 families with their associated soluble cytokine receptors. High values indicated a lower survival rate. This prediction was to some extent better than the standard clinical predictors used. The results highlight potential new targets for immune treatment development. Additionally, these findings could contribute to the refinement of cancer treatment strategies, particularly in the context of emerging immune therapies.

**Abstract:**

Activation of the acute-phase cascade (APC) has been correlated with outcomes in various cancers, including head and neck squamous cell carcinoma (HNSCC). Primary drivers of the APC are the cytokines within the interleukin-6 (IL-6) and IL-1 families. Plasma levels of IL-6 family cytokines/soluble receptors (IL-6, IL-27, IL-31, OSM, CNTF, soluble (s-)gp130, s-IL-6Rα) and IL-1 family members (IL-1RA, s-IL-33Rα) were determined at diagnosis for 87 human papillomavirus (HPV)-negative (−) HNSCC patients. We then studied the 5-year Disease-Specific Survival (DSS) and Overall Survival (OS). Increased plasma levels of IL-6 (*p* < 0.001/*p* < 0.001) (DSS/OS), IL-31 (*p* = 0.044/*p* = 0.07), IL-1RA (*p* = 0.004/*p* = 0.035), soluble (s)-IL-6Rα *p* = 0.022/*p* = 0.035), and s-gp130 (*p* = 0.007/*p* = 0.003) at diagnosis were predictors of both OS and DSS from HPV(−) HNSCC patients. The cytokine DSS/OS predictions were associated with TNM stage and smoking history, whereas the soluble receptors IL-6Rα, gp130, and IL33Rα more uniquely predicted DSS/OS. Clinically, IL-6 levels above 2.5 pg/mL yielded 75% specificity and 70% sensitivity for DSS. In conclusion, high plasma levels of IL-6, IL-31, and IL-1RA, as well as the soluble receptors IL-6Rα, gp130, and IL33Rα, predicted clinical outcome. This shows their potential as candidates for both general therapy and immune therapy stratification, as well as being future platforms for the development of new immunotherapy.

## 1. Introduction

In 2020, over 900,000 new cases of head and neck cancer (HNC) (lip, oral cavity, pharynx, larynx, or salivary glands) were estimated worldwide [1,2]. Approximately 50% of the newly diagnosed HNC patients are alive at 5 years [1]. The pathogenesis of HN squamous cell carcinoma (HNSCC) is related to mucosal inflammation caused by local irritants such as tobacco, alcohol, and betel nuts [3], or by the human papilloma virus (HPV) [4]. The type and localization of inflammation are clearly related to the clinical and biological characteristics of HNSCC. HPV-positive (+) tumor patients demonstrate superior survival rates compared to HPV(−) tumor patients when originating from the oropharynx (OP) [5]. 

The general acute-phase inflammatory cascade (APC) includes proteins that are classically primary responders to bacterial infections or tissue necrosis [6,7]. The APC response is seen at the cellular, intercellular, and systemic level. Prominent members of the APC include C-reactive protein (CRP) [8], cytokines in the interleukin (IL)-6 family, and cytokines in the IL-1 family [6,9]. Unlike other cytokines that primarily exert local effects, these APC players also exhibit hormone-like effects on distant organ systems [10,11]. This is most prominent for IL-6, where, for example, systemic inflammation markers such as CRP, leukocytosis, and thrombocytosis are closely related secondary actions associated with its activity. Furthermore, the systemic inflammatory response, including the systemic levels of acute-phase cytokines, has an independent prognostic impact in several cancers, including HNSCC [12,13,14,15,16,17,18,19,20,21,22,23,24,25,26,27]. 

IL-6 is a pleiotropic hormone that plays an important role in tissue homeostasis in the bone marrow, liver cells, and mucosa [28,29,30], as well as being a driver of acute and chronic inflammation [31]. The pleiotropic effects of IL-6 can be explained through the various modes of IL-6 signaling: classical IL-6 signaling, IL-6 *trans*-signaling, and IL-6 cluster signaling [11]. In classic signaling, IL-6 binds directly to the membrane-bound IL6Rα–gp130 receptor complex, which only occurs on cells expressing the IL-6 receptor, such as hepatocytes, neutrophils, monocytes, macrophages, and some lymphocytes [10]. In IL-6 *trans*-signaling, IL-6 binds soluble IL6Rα (s-IL6Rα) and then binds to membrane-bound gp130 [32,33]. Cluster signaling involves complexes of membrane-bound IL-6–IL-6R on one cell, thus activating gp130 subunits on target cells [34]. *Trans*-signaling may also be blocked by a soluble dimeric version of gp130 (s-gp130) that specifically inhibits the pro-inflammatory effects of IL-6 [35]. Furthermore, the IL6 cytokine family [10,36] includes IL-11, IL-27, IL-31, ciliary neurotrophic factor (CNTF), leukemia inhibitory factor (LIF), oncostatin M (OSM), and cardiotrophin-like cytokine factor 1 (CLC) [37]. These cytokines share gp130 as their signal transducer but also mediate cytokine-specific effects through a ligand-specific receptor [37]. Mostly, the associations of these cytokines with HNSCC are limited. 

However, based on this present cohort, we have previously suggested that IL-6, but also IL-31 levels, predicts DSS [38]. Furthermore, the soluble cytokine receptor IL-6Rα(s-IL-6Rα) and the associated soluble gp130 (s-gp130) showed a trend towards the same prediction. We have previously studied DSS predictions from clustering analyses [38]. Indeed, three clusters could be established with different survival predictions. Presently, we report from an extended observation studying single cytokines/soluble cytokine receptors, including a full five-year analysis of DSS and five-year OS. OS analyses decrease biased election of patients of DSS versus other death causes, thereby increasing the strength of the results. Furthermore, inflammatory cytokines also prognostically relate to other cancers [12,13,14,15,16,17,18,19,20,21,22,23,24,25,26,27], as well as cardiovascular disease [39], and thus studying OS broadens the potential to deepen our understanding of APC biology. IL-6 family cytokines are often secreted together with IL-1 family cytokines during inflammation [40]. We have therefore included IL-1RA and s-IL-33Rα/ST2 in this study.

Immunotherapy has been introduced as a promising treatment option for HNSCC patients [41]. Unfortunately, not all patients benefit from such treatment. Plasma levels of interleukins and associated molecules can both yield prognostic information in general and may help identify patients who are most likely to benefit from immunotherapy [42]. Thus, there is considerable interest in establishing new immunologically derived cytokine candidates to refine patient selection for established immune-based treatments. Moreover, proposing new candidates for development of such therapies holds promise in advancing personalized cancer treatment. 

The HPV(+) individuals presently had more than a 90% 5-year DSS, leaving the 5-year survival rate to be meaningfully studied only among the HPV(−) patients. Thus, it is reported from the HPV(−) patients only in the current study. We investigated the interaction between the APC predictions and conventional clinical predictors of HNSCC survival. We analyzed the five-year survival predictions presented in this study, adjusting for patient age at diagnosis, as well as TNM stage and reported smoking consumption. From the HPV(−) patients, we demonstrated, for example, that a single blood sample can be utilized to identify patient groups with different prognoses based on changed cytokine/cytokine-receptors levels. For example, one group consisted of patients with a below-median IL-6 level, achieving close to 100% 5-year Disease-Specific Survival (DSS). Another group comprised eight patients with high s-gp130 and extended disease, where only one out of eight patients achieved a five-year DSS survival.

## 2. Materials and Methods

### 2.1. Patients Included in the Study

The study included 87 HPV(−) patients selected from 144 consecutive patients in the original cohort of newly diagnosed HNSCC based on tumor HPV presence or not. The patients were admitted to the Department of Otolaryngology and Head and Neck Surgery, Haukeland University Hospital (Bergen, Norway), from 2013 to 2018. Only patients treated with curative intent were included. Patients with autoimmune disease or on systemic corticosteroid therapy were excluded from the study. The HPV(−) patient characteristics are given in Table 1. The five-year survival by the end of 2023 was determined from the Norwegian Population Registry.

### 2.2. Laboratory Analyses

The biopsy analyses for human papilloma virus (HPV) infection have been previously described in detail [43,44]. 

Peripheral venous blood samples were collected during initial hospital admission for HNSCC treatment. Samples were stored at room temperature for less than two hours before undergoing centrifugation and collection of plasma. The plasma was prepared by gradient centrifugation with Lymphoprep^®^ (Nycomed Pharma AS, Oslo, Norway) aliquoted and stored at −80 °C until analysis. 

The samples were thawed and centrifuged at 16,000× *g* for 4 min before analysis. The levels of s-IL-33Rα and IL-1RA were analyzed using Luminex analyses (R&D Systems Europe Ltd., Abingdon, UK). IL-6 levels were determined using a high-sensitivity IL-6 Luminex kit (Invitrogen, Biosource, Carlsbad, CA, USA), while IL-1β levels were determined using a High-Sensitivity ELISA kit (R&D Systems). IL-1β, sgp130, IL-6Rα, IL-27, IL-31, OSM, CNTF, and TNF-α were determined using the Human Premixed Multi-Analyte Kit for Luminex technology (R&D Systems), whereas CNTF was analyzed with a kit from EMD Millipore Corporation, Billerica, MA, USA. All analyses were performed strictly according to the manufacturer’s instructions using a Luminex^®^ 100^TM^ (Luminex Corporation, Austin, TX, USA). All results are presented as the mean level of duplicate determinations. One of the patient samples was included in all assays to evaluate inter-platelet variation. No substantial differences were detected between the different assays. The variation between the duplicates was generally <10% of the mean concentration. Cytokine/soluble cytokine receptors levels showed no significant correlations with sample storage time. For one HPV(−) patient, we only had clinical data and IL-6 values available for analysis. 

### 2.3. Tobacco History Scores

From the hospital records, information about patient smoking history was obtained and scored based on pack years, defined as 20 cigarettes daily for one year. The cumulative scoring was as follows: 0 = never smoked, 1 = less than 10 pack years, 2 = probably less than 10 pack years, 3 = probably more than 10 pack years, and 4 = more than 10 pack years. The smoking status information for two of the HPV(−) patients was not available.

### 2.4. Statistical Analyses

The IBM^®^ SPSS^®^ Statistics software, version 29.0 (IBM Corp., Armonk, NY, USA), was employed. Categorized descriptive data were compared by cross-tables and the exact chi-squared test. The Mann–Whitney U test was utilized for comparing different groups, and Pearson’s correlations were employed for correlation analyses. Receiver Operating Characteristic (ROC) analysis was used to estimate sensitivity and specificity with cut-off values in predicting IL-6 levels for Disease-Specific Survival (DSS). Kaplan–Meier analyses were conducted to estimate survival curves, and the Log-Rank test was applied to compare survival between groups. Multivariate Cox proportional hazard models were used to analyze the effect of introducing covariates to cytokine/soluble cytokine receptor survival predictions. The results are reported as the relative risk (RR) with a 95% confidence interval. The cytokines/receptors were either dichotomized by the median (IL-6) or quartiles of the entire cohort (Appendix A). The age of the patients at diagnosis was incorporated into the analyses in the form of quartiles. TNM scores were staged according to the 7th TNM edition of classification of malignant tumors by the International Union Against Cancer. TNM stage and smoking history were scored as originally recorded. The T and N Integer Stage (TANIS) was derived based on the TNM stage, calculated as the numerical sum of the T and N stages. All analyses were performed as two-tailed analyses. A two-tailed *p* value < 0.05 was considered statistically significant. Final figures were created using R statistical software (version 4.2.3, R Core Team, Vienna, Austria) with various packages, including but not limited to pROC (version 1.18.0), survival (version 3.5.7), survminer (version 0.4.9), and ggplot2 (version 3.4.0). 

## 3. Results

### 3.1. Patients Included in the Study

The HPV(+) patients exhibited a notably lower incidence of disease-specific deaths, with only 3 disease-specific and 4 overall deaths, in contrast to 20 HNSCC deaths and 6 non-index HNSCC deaths observed in the HPV(−) group. This discrepancy is visually represented in Figure 1, which depicts the five-year DSS from HPV(+) compared to HPV(−) patients. Interestingly, among the HPV(+) patients, only the cytokine/cytokine receptor IL-6 predicted survival measured at the 5-year mark. Consequently, only consecutive HPV(−) patients (N = 87) were included in these analyses, selected from the original cohort of 144 patients (Table 1 and Appendix A). Fifty-seven of the original patient cohort (40%) had HPV(+) tumors. Forty-three patients were not subjected to testing but were labeled HPV(−) as they had primary tumors outside the oropharynx area [45]. The median age at diagnosis was 66 years, ranging from 37 to 82 years. Sixty-eight percent (N = 59) of patients were male (Table 1).

#### 3.1.1. Five-Year Survival among HPV(+) Patients

We analyzed both the 5-year DSS and OS among the HPV(+) patients, considering the studied cytokines and soluble cytokine receptors. None of these values predicted DSS or OS, except for the IL-6 values measured binomially. Therefore, these data are not reported.

#### 3.1.2. Correlation between Cytokines/Soluble Receptors, Age, TNM Stage, Smoking History and CRP Levels

A significant correlation was observed between IL-6 levels and age (r = 0.32; *p* < 0.01), T stage (r = 0.451; *p* < 0.01), N stage (r = 0.29; *p* < 0.01), and smoking history (r = 0.324; *p* < 0.01), while IL-31 levels correlated with N stage (r = 0.274; *p* < 0.05). IL1RA levels correlated with N stage (r = 0.300; *p* < 0.01) and tobacco history (r = 0.235; *p* < 0.05). Lastly, ST2/s-IL33Rα levels correlated with tobacco history (r = 0.235; *p* < 0.05). CRP levels correlated with T stage (r = 0.31, (*p* < 0.001), IL-6 levels (r = 0.43; *p* < 0.001), and s-IL-33Rα (r =0.31; *p* < 0.01.

#### 3.1.3. DSS from Clinical Parameters

As expected, HPV(+) patients exhibited better survival outcomes compared to their HPV(−) counterparts within the entire cohort (Figure 1). Advanced T stage was associated with inferior 5-year DSS (*p* < 0.001). The same was observed for N stage (*p* < 0.001). Increased levels of smoking exhibited a noticeable trend towards decreased survival, but statistical significance in relation to the 5-year DSS was not reached (*p* = 0.077) (Table 1). 

#### 3.1.4. Individual Acute-Phase Cytokines and Soluble Cytokine Receptors Are Associated with Prognosis in HNSCC

By analyzing survival using the Kaplan–Meier estimates and stratifying patients based on IL-6 levels (quartiles), inferior 5-year DSS (Figure 2A) (*p* < 0.001) and OS (Figure 2B) (*p* < 0.001) was observed for the patients. Significant 5-year DSS (*p* < 0.001) (Figure 2C) and OS (*p* < 0.001) (Figure 2D) differences in survival were also observed when using the median value of IL-6 to dichotomize the survival curves. IL-31 levels also predicted the 5-year DSS (*p* < 0.03) (Figure 3A) as well as a trend towards predicting OS (*p* < 0.08) (Figure 4A). Level of s-IL-6Rα predicted the DSS (*p* < 0.04) (Figure 3C) and OS (*p* < 0.04) (Figure 4C). Furthermore, when examining s-gp130, high levels were associated with a worse 5-year DSS (*p* < 0.02) (Figure 3D) and OS (*p* < 0.003) (Figure 4D). IL-1RA also predicted the 5-year DSS (*p* = 0.005) (Figure 3B) and OS (*p* = 0.001) (Figure 4B). The other studied cytokines/cytokine receptors did not predict survival in the HPV(−) patients, as indicated by the Kaplan–Meier analysis. Notably, for all the prediction results, higher values in the HPV(−) cohort were consistently associated with a worse prognosis (see Figure 3 and Figure 4).

#### 3.1.5. Prognostic Five-Year Survival Analyses by Individual Cytokine and Soluble Cytokine Receptors Using Cox Regression Analysis 

Multivariate Cox regression survival analyses were conducted, incorporating gender, age, T and N stage, and smoking history, along with one of the indicated cytokines/soluble receptors (Table 2). Concerning IL-6, IL-31, and IL-1RA, the mentioned covariates accounted for the survival predictions (Table 2). When the soluble cytokine receptors were introduced into the analysis, s-gp130 continued to demonstrate a significant 5-year DSS (RR = 1.74; CI 1.16–2.61, *p* = 0.008) and 5-year OS (RR = 1.53, CI 1.12–2.08, *p* = 0.007). Similarly, s-IL-33Rα demonstrated a significant 5-year DSS (RR = 1.79, CI 1.19–2.69, *p* = 0.005), whereas s-IL-6Rα showed a trend in DSS (RR = 1.40; CI 0.97–2.02, *p* = 0.068) and a significant OS prediction (RR = 1.36, CI 1.02–1.81, *p* = 0.039) (Table 2).

#### 3.1.6. Receiver Operating Characteristic (ROC) Curve Analysis

Receiver Operating Characteristic (ROC) analysis was conducted to determine a cut-off level for our measured IL-6, which can be clinically employed to estimate the likelihood of a 5-year DSS at the individual level. The ROC curve for the HPV(−) patients is presented in Figure 5. The area under the curve was 0.77 with a 95% confidence interval (CI) of 0.65–0.89 (*p* < 0.001). A cut-off of IL-6 = 2.5 pg/mL provided 70% sensitivity and 75% specificity in our study.

#### 3.1.7. Cytokine-Predicted Survival by IL-6 Level Adjustment

Survival analysis based on above-median IL-6 values is presented in Figure 6. In the low IL-6 value group, a noticeable trend in Disease-Specific Survival is observed. When analyzing high IL-6 values in relation to the quartiles of IL-31, a significant survival prediction is evident, with a *p* value of 0.043 for the DSS (Figure 6D). Similarly, for IL1-RA, a *p* value of 0.029 signified a significant 5-year Disease-Specific Survival and Overall Survival (*p* < 0.02) (Figure 6B,E). Soluble gp130 values maintained a noticeable trend for the 5-year DSS (Figure 6C) but a significant prediction for the 5-year OS (*p* < 0.04) (Figure 6F).

#### 3.1.8. Cytokine-Predicted Survival Using TANIS Adjustment

Survival analysis using the median TANIS (T stage + N stage sum (integer) score) [46] revealed distinct outcomes in the two groups of patients based on their IL-1RA and s-gp130 adjusted values. Notably, the extended TANIS group exhibited a higher incidence of disease-caused deaths, as anticipated (Figure 7). Specifically, within the subgroup characterized by both high TANIS and high s-gp130 values, 7 out of 8 patients died of HNSCC, contrasting with 11 out of 25 patients in the extended TANIS stage and low s-gp130 subgroup (*p* < 0.004) (Figure 7B). Conversely, in patients with extended TANIS, there is a noticeable trend, whereby none of the 22 patients with low IL1RA experienced disease-specific death, compared to 18 of 30 patients who died with higher IL-1RA levels (*p* = 0.073 combined) (Figure 7C,D). Additional analyses of the Overall Survival (OS) demonstrated a significant predictive value (*p* = 0.011 combined).

## 4. Discussion

In the present study, we investigated the cascade of acute-phase cytokine levels of IL-6 and IL-1 family cytokines and associated soluble receptors at time of diagnosis in relation to the 5-year survival in a cohort of 87 HNSCC HPV(−) patients. Regarding the IL-6 family, high values of IL-6 and IL-31, along with the associated soluble receptors IL-6Rα and gp130, predicted worse OS and DSS. Increased IL-1RA also predicted lowered OS and DSS. The IL-6, IL-31, and IL-1RA predictions were primarily associated with tumor extent and smoking history, whereas the soluble receptors (s)-IL-6Rα, s-gp130, and s-IL-33Ra predicted survival even after such adjustments.

The present results pertain primarily to the HPV(−) patients, as the number of deaths observed among the HPV(+) patients was too low to draw meaningful conclusions. To comprehensively study this subject matter, a larger sample size with a longer observation period is necessary. Further research with an expanded patient population and extended follow-up duration is warranted to provide more conclusive insights, particularly regarding HPV(+) patients.

High IL-6 levels have been shown to be associated with a poorer prognosis in cancer, HNSCC included [13,20,21,22,23,24,25,26,27]. In this article, we have confirmed and extended this finding to HNSCC. IL-6 is a member of the IL-6 cytokine family, i.e., several cytokines with common gp130 receptor use [6,32]. IL-31, a member of the IL-6 family, also predicted the 5-year DSS and showed a trend towards OS prediction. To the best of our knowledge, this is the first published cohort of HNSCC patients showing a significant survival prediction when applying this cytokine. While the other studied IL-6 family cytokines (IL-27, OSM, CNTF) did not predict survival, the s-IL-6Rα and s-gp130 receptors, which are associated with this family, currently demonstrated prediction for DSS and OS. These findings, to the best of our knowledge, are also shown for the first from this cohort.

As high IL-6, s-IL-6Rα, and s-gp130 predicted survival among the HPV(−) tumor patients, activation of both the classical and *trans* intracellular pathways seem to be associated with both increased HNSCC-caused and general mortality. Paradoxically, s-gp130 is considered a member of the inhibitory system [47], and as such, this finding warrants further investigation. 

In addition to IL-6 family members, other cytokines are viewed as secondary drivers of the APC [6], and for this reason we have also included the IL-1 family inhibitor IL-1RA as well as the s-IL-33Rα. We found that high IL-1RA levels predicted a decreased 5-year survival. IL-1RA is an acute-phase marker, and high levels have previously been associated with aggressive disease and/or unfavorable prognosis in various malignancies, including T-cell lymphoma [48], sarcoma [49], colorectal cancer [50,51], and thyroid cancer [52]. Our study is, to the best to our knowledge, the first to suggest that this also holds true for HNSCC patients.

The levels of these inflammatory markers can also be influenced by inflammageing, i.e., a smoldering of the inflammatory system that can be a part of the aging process [53]. Therefore, we adjusted by the age of the patients. This adjustment did not alter the survival predictions. Disease extent, as measured by the TNM stage, presently predicted HNSCC survival [54]. Smoking causes HNSCC and is also an inflammatory driver [55]. The rate of smoking is furthermore tied to a worse survival rate [56]. Consequently, we conducted Cox regression analyses, considering the smoking level and TNM stage as covariates, to explore the cytokine survival predictions. The 5-year cytokine predictions were largely statistically secondary to a combination of TNM stage and smoking history. On the other hand, high levels of the studied soluble interleukin receptors still predicted survival even after adjusting for age, gender, TNM stage, and smoking history. 

The present observations show that the HNSCC patient cytokine and cytokine receptor levels likely reflect separate aspects of the cancer-associated systemic acute-phase cytokine response, with one dimension associated with the cytokines IL-6, IL-31, and IL-1RA, whereas another dimension is associated with the soluble receptors IL-6Rα, gp130 levels, and IL-33Rα/ST2. One explanation may be that highly malignant HNSCC tumors have a higher overall cellular turnover than their more benign counterparts [57]. The present results could then be explained by more receptor shedding from highly malignant tumors. 

These present findings underline the importance of IL-6-caused *trans* activation as an important part of HNSCC tumor biology. The relatively narrow range of cells that can be stimulated by *cis* activation, compared to *trans* activation [58], broadens the possible checkpoints where the inflammatory and cancer systems may interact. This study supports the idea that these phenotypic inflammation characteristics reflect different underlying drivers, as observed in other comparable phenomena [5].

We only included patients primarily treated with curative intent. The inclusion of newly diagnosed palliative or even cachectic patients would have incorporated individuals with substantial inflammation present at diagnosis [59]. The results could have then reflected cancer end-stage inflammation rather than an HNSCC-specific relation. As a result, our findings are more applicable to the HNSCC disease in broader terms.

The present primary cohort predominantly consisted of oral cavity SCC patients in the HPV(−) group. An intriguing aspect is the suggestion that IL-6 plays a significant role in oral cancer by activating the STAT3 pathway. This activation influences anti-apoptotic genes as well as autocrine stimulation of IL-6 secretion [60], underscoring the importance of the IL-6 concentration. As such, it has recently been suggested in a review paper that inflammatory markers, including IL6, may be used as predictors of the recurrence of oral head and neck carcinomas [61].

Only one blood sample was drawn from each patient. The half-life of the studied cytokines in blood is unknown, and the blood cytokine concentration may vary substantially over time. However, we have previously conducted a study with renal carcinoma patients demonstrating the cytokines levels in blood to be surprisingly stable [40], supporting the validity of the present methodical approach.

As previously mentioned, the inflammatory system plays a crucial role in cardiovascular diseases [39], with IL-6 being a major factor in the process [62]. The extent to which inflammation is related to cardiovascular diseases and cancer crosstalk is of interest and warrants more extensive study. 

The importance of the immune system in patient prognosis encourages further explorations of immune-modulating treatments for HNSCC patients [63]. The current results suggest targeting the inflammatory dimensions of the HNSCC disease. Additional research into the interactions between carcinomas and the immune system in a broader context is needed. 

Clinically, those with the lowest half of IL-6 values had close to 100% survival. We furthermore conducted an IL-6-value-based ROC analysis in our HPV(−) HNSCC sub-cohort. A cut-off at 2.5 pg/mL in our assay provided both sensitivity and specificity of more than 70% for the risk of disease-specific death. Such information could be used to further personalize HNC treatment. Additionally, in a more individualized analysis that excludes patients with other explanatory causes for high IL-6 values, one could even potentially improve this prognostic factor [64]. Furthermore, high gp130 showed a DSS rate of one in eight among patients with advanced disease (i.e., high T+N stage). This shows the need to improve prognosis prediction beyond the TNM stage. While smoking information might offer some such insights, its accuracy relies on patient reporting, which may not always be 100% correct [65]. Therefore, easily available laboratory immune parameters should be better suited for such purposes. 

Several investigators have shown that immune-based therapy may provide an excellent effect even in a palliative setting [66]. If a serious prognosis is given, determined at diagnosis by extended clinical disease and supported by the presence of identified immune activation [67], there is a tempting prospect of extending treatment with (neo)adjuvant immunotherapy in future trials. Supporting this idea, it has been demonstrated that IL-6 levels may predict immunotherapy response in other carcinomas [42]. As presently shown, elevated values of additional cytokines and soluble receptors beyond IL-6 could serve as supplementary biomarkers with the goal of identifying such suitable patients. Administering double-adjuvant immunotherapy, despite the current high frequency of side-effects associated with such treatment [68], could then be an excellent therapeutic option.

## 5. Conclusions

In conclusion, increased IL-6, IL-31, IL-1RA, s-IL-6Rα, and s-gp130 in the plasma predicted poorer HNSCC 5-year DSS and OS, as evidenced by the survival analyses. Our findings support the idea that cytokines reflect one HNSCC inflammatory dimension, whereas the soluble receptors IL-33Rα/ST2, gp130, and IL-6Rα represent another distinct HNSCC inflammatory dimension. We advocate for further research into the soluble cytokine receptors specifically studied in our cohort of HNSCC patients. The prognostic implications of IL-6, IL-1RA, and s-gp130 levels extend beyond the TNM-generated cancer prognosis, offering potential clinical applications.

## Figures and Tables

**Figure 1 cancers-16-01484-f001:**
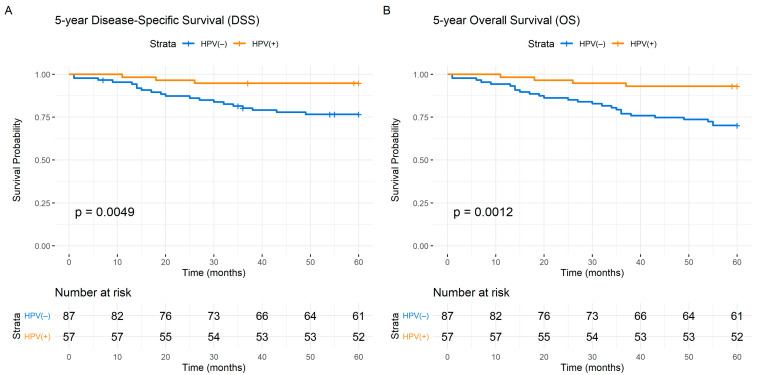
Kaplan–Meier survival curves showing the 5-year Disease-Specific Survival (DSS) and 5-year Overall Survival (OS) for the entire cohort by HPV status. (**A**): Disease-Specific Survival for the entire cohort (N = 144), HPV(−) (blue), and HPV(+) (orange). (**B**): Overall Survival (OS) for the entire cohort (N = 144), HPV(−) (blue), and HPV(+) (orange). The numbers at risk are indicated at the bottom of the graph. The *Y*-axis represents the probability of survival and *X*-axis denotes time in months. Differences between the groups were examined with Log-Rank tests and presented with the associated *p* values.

**Figure 2 cancers-16-01484-f002:**
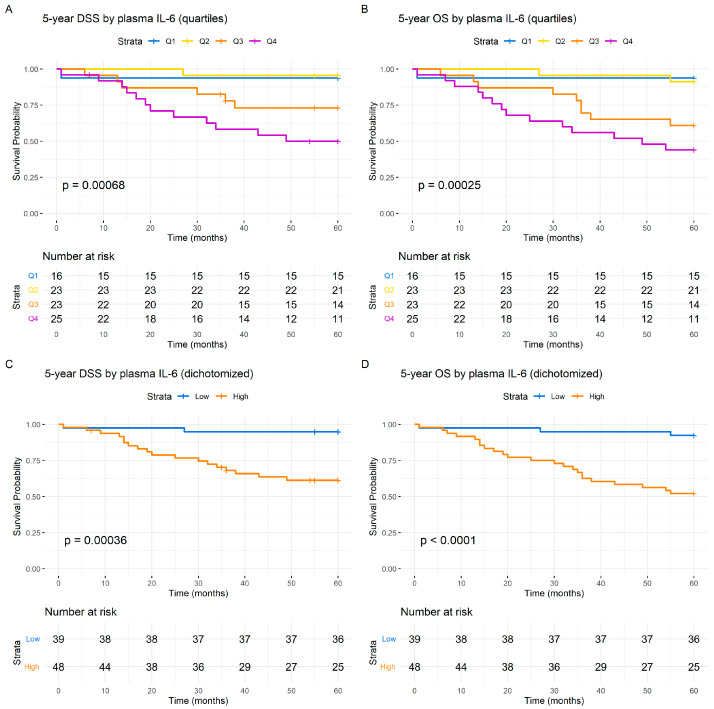
Kaplan–Meier prognosis from the pretreatment of IL-6 levels on the 5-year Disease-Specific Survival (DSS) and 5-year Overall Survival (OS) by quartiles and median values for the HPV(−) cohort. (**A**). 5-year DSS. IL-6 levels are presented as quartiles with Q1 (blue), Q2 (yellow), Q3 (orange), and Q4 (purple). (**B**). 5-year OS. IL-6 levels are presented as quartiles with Q1 (blue), Q2 (yellow), Q3 (orange), and Q4 (purple). (**C**). 5-year DSS. IL-6 levels are presented as a binomial variable with above (orange) or below median (blue) levels. (**D**). 5-year OS. IL-6 levels are presented as a binomial variable with above (orange) or below median (blue) levels. The numbers at risk are indicated at the bottom of the graph. The *Y*-axis represents the probability of survival and the *X*-axis denotes the time in months. The differences between the groups were examined with Log-Rank tests and presented with the associated *p* values.

**Figure 3 cancers-16-01484-f003:**
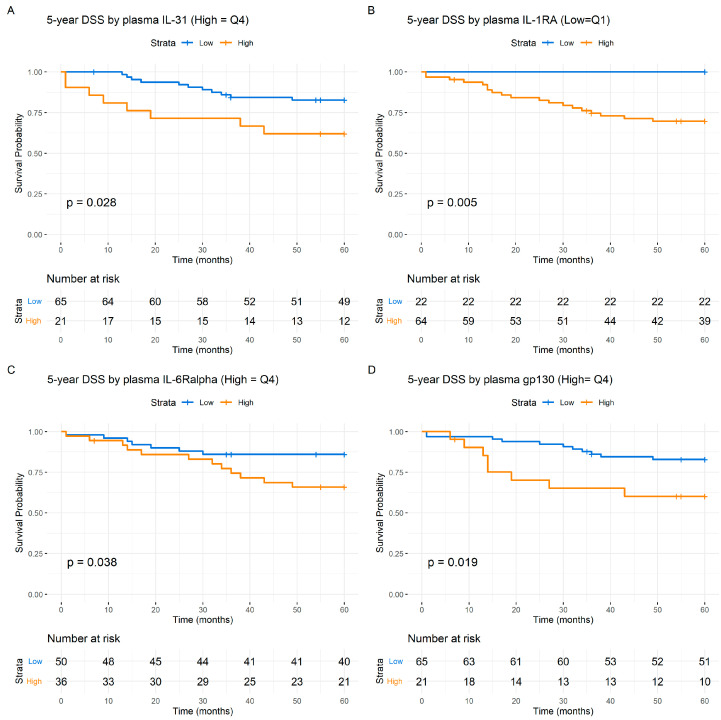
Kaplan–Meier 5-year Disease-Specific Survival (DSS) from the pretreatment of cytokine/soluble receptor levels in the HPV(−) patients: blue, low value; orange, high value. (**A**), IL-31; (**B**), IL-1RA; (**C**), IL-6Rα; (**D**), gp130. The numbers at risk are indicated at the bottom of the graph. The *Y*-axis represents the probability of survival and the *X*-axis denotes the time in months. The differences between the groups were examined with Log-Rank tests and presented with the associated *p* values. HPV(+) patients showed no significant DSS dependent on the indicated cytokine/cytokine receptor values.

**Figure 4 cancers-16-01484-f004:**
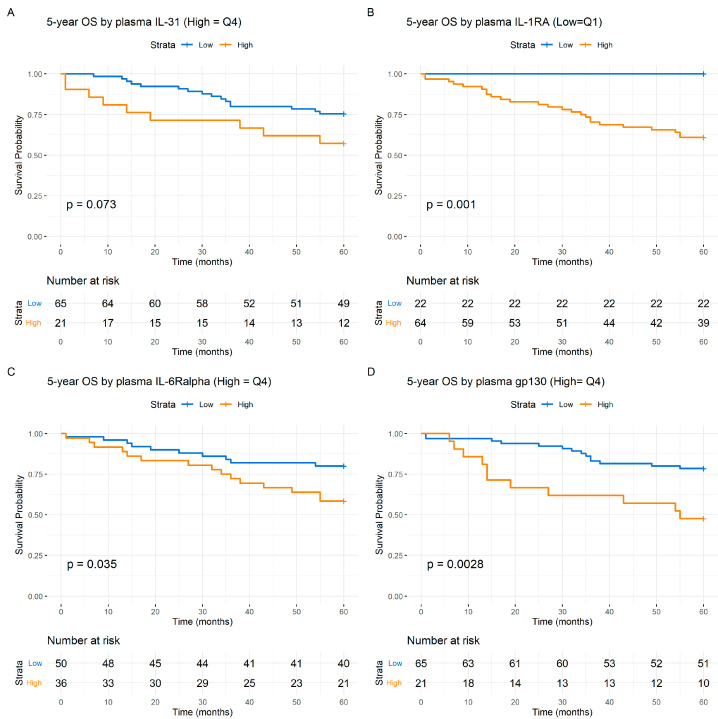
Kaplan–Meier 5-year Overall Survival (OS) from the HPV(−) patients. Pretreatment levels of cytokines: blue, low value; orange, high value. (**A**), IL-31; (**B**), IL-1RA; (**C**), IL-6Rα; (**D**), gp130. The numbers at risk are indicated at the bottom of the graph. The *Y*-axis represents the probability of survival and the *X*-axis denotes the time in months. The differences between the groups were examined with Log-Rank tests and presented with the associated *p* values.

**Figure 5 cancers-16-01484-f005:**
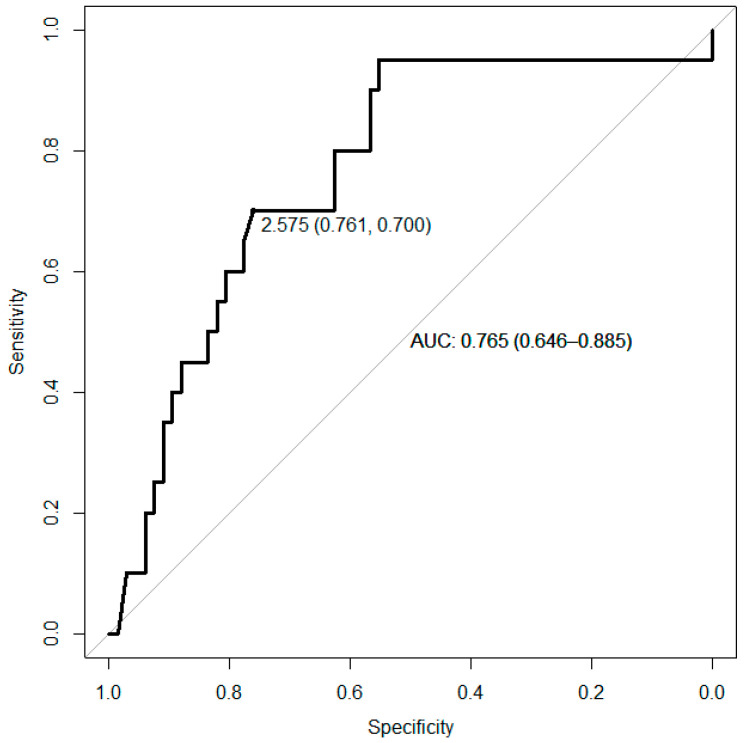
Receiver Operating Characteristic (ROC) curve studying the measured IL-6 levels for the 5-year Disease-Specific Survival (DSS) for HPV(−) patients. The area under the curve (AUC) was 0.765 with a 95% confidence interval (CI) of 0.65–0.89 (*p* < 0.001). Patients with a cut-off at 2.5 pg/mL exhibited 70% sensitivity and 75% specificity. The *X*-axis represents specificity and the *Y*-axis represents sensitivity.

**Figure 6 cancers-16-01484-f006:**
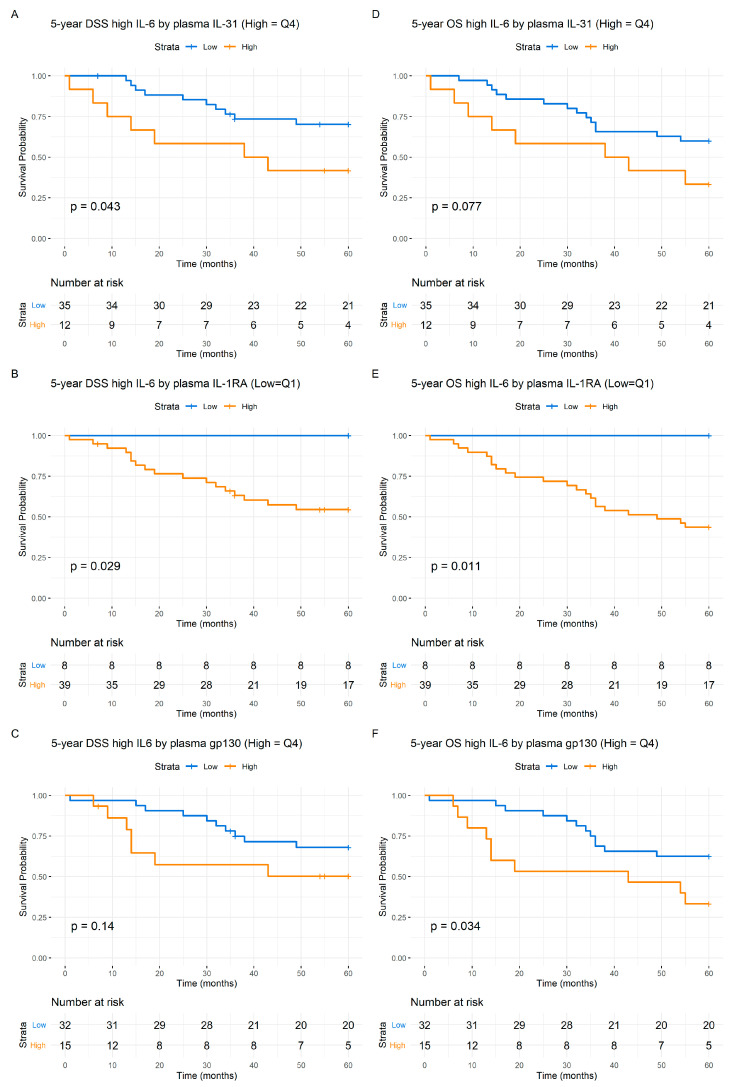
Kaplan–Meier survival curves depicting the 5-year Disease-Specific Survival (DSS) and Overall Survival (OS) concerning above-median levels of IL-6 and other respective cytokines. Low cytokine values are represented in blue, while high levels are depicted in orange. Panels (**A**–**C**) show the DSS, while panels (**D**–**F**) represent the OS. Each set of panels corresponds to quartiles of the different variables: panels (**A**,**D**) with respect to quartiles of IL-31; panels (**B**,**E**) with respect to quartiles of IL1-RA; and finally, panels (**C**,**F**) with quartiles of s-gp130. The numbers at risk are shown at each time point. Statistical comparisons were performed between the high and low quartiles and combined using Log-Rank tests and the associated *p* values. The *Y*-axis represents the probability of survival and the *X*-axis denotes the time in months.

**Figure 7 cancers-16-01484-f007:**
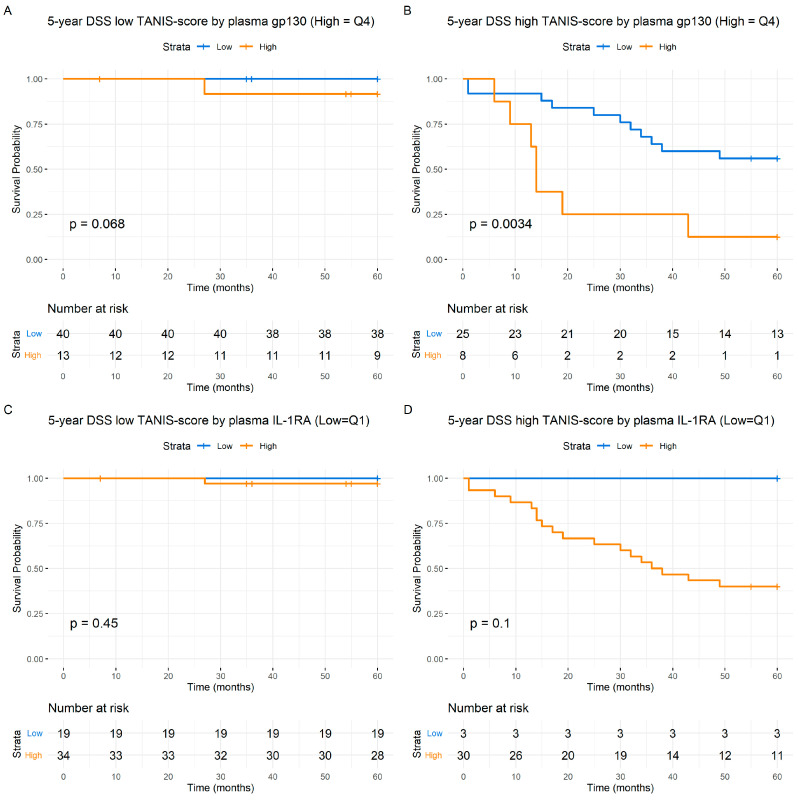
Kaplan–Meier survival curves depicting the 5-year Disease-Specific Survival (DSS) with low (**A**,**C**) and high median (**B**,**D**) TANIS scores (T stage+N stage sum (integer) score). Panels (**A**,**B**) compare the impact of TANIS on survival with respect to plasma s-gp130, depicting low (**A**) and high (**B**) TANIS scores. High values of s-gp130 are shown in orange, while low values are in blue. Panels (**C**,**D**) explore the relationship between TANIS and IL1-RA, showcasing low (**C**) and high (**D**) TANIS-score cohorts. High values of IL1-RA are shown in orange, while low values are in blue. The numbers at risk are shown at each time point. Statistical comparisons were performed between the high and low quartiles and combined using Log-Rank tests and the associated *p* values. The *Y*-axis represents the probability of survival and the *X*-axis denotes the time in months.

**Table 1 cancers-16-01484-t001:** Clinical HPV(−) patient characteristics at diagnosis.

Variable		Distribution	Prediction of 5-Year DSS
Age at diagnosis	Years (Mean ± SD)	62.9 ± 10.3	n.s.
Gender	Males	59	n.s.
	Females	28	
T stage	0	3	*p* < 0.001
	1	29	
	2	34	
	3	8	
	4	13	
N stage	0	57	*p* < 0.001
	1	9	
	2	20	
	3	1	
Smoking	Never smoked	21	*p* = 0.077
	<10 pack years	1	
	Probably <10 pack years	8	
	Probably >10 pack years	5	
	>10 pack years	50	
	Missing	2	
Site (ICD-10)	Oropharynx	17	
	Oral Cavity	61	
	Other	9	
Total patients		87	

DSS: Disease-Specific Survival analysis by Kaplan–Meier methods with Log-Rank comparison between groups. n.s.: not significant.

**Table 2 cancers-16-01484-t002:** Cox multivariate regression analyses investigating the 5-year DSS/OS, adjusting for gender, age, T and N stage, and tobacco-smoking history (5-step). Each indicated cytokine/soluble cytokine receptor was analyzed separately. For soluble gp130, all adjusting factors are presented, while for the remaining, only their specific contribution is shown. All cytokine/cytokine receptor values were analyzed binomially based on quartiles.

	Covariate	Outcome	*p* Value	RR (95% CI)
Block I + s-gp130	Gender	DSS	0.279	0.50 (0.15–1.74)
	Age		0.203	0.96 (0.90–1.02)
	T stage		<0.001	2.46 (1.52–3.98)
	N stage		<0.001	5.28 (2.59–10.8)
	Tobacco		0.111	1.39 (0.93–2.05)
	s-gp130		0.008	1.74 (1.16–2.61)
	Gender	OS	0.359	0.63 (0.24–1.69)
	Age		0.928	1.00 (0.95–1.06)
	T stage		0.006	1.69 (1.17–2.46)
	N stage		<0.001	3.40 (2.08–5.55)
	Tobacco		0.143	1.27 (0.92–1.73)
	s-gp130		0.009	1.53 (1.12–2.08)
Block I + s-IL-6Rα	s-IL-6Rα	DSS	0.005	1.40 (0.97–2.02)
	s-IL-6Rα	OS	0.016	1.36 (1.02–1.81)
Block I + s-IL33Rα/ST2	s-IL33Rα/ST2	DSS	0.005	1.79 (1.19–2.69)
	s-IL33Rα/ST2	OS	0.016	1.48 (1.08–2.03)
Block I + IL1-RA	IL1-RA	DSS	0.966	-
	IL1-RA	OS	0.955	-
Block I + IL-6	IL-6	DSS	0.667	-
	IL-6	OS	0.876	-
Block I + IL-31	IL-31	DSS	0.138	-
	IL-31	OS	0.685	-

## Data Availability

The datasets presented in this article are not allowed to be shared due to national legal regulations.

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
