# Peer review of "Plasma IL-1 and IL-6 Family Cytokines with Soluble Receptor Levels at Diagnosis in Head and Neck Squamous Cell Carcinoma: High Levels Predict Decreased Five-Year Disease-Specific and Overall Survival"

_cancers, 2024, doi:10.3390/cancers16081484_

Round 1

Reviewer 1 Report

Comments and Suggestions for Authors

1.      Summary

The manuscript by Aarstad and colleagues evaluates the potential prognostic value of plasma cytokine levels and soluble receptors in predicting Disease-Specific Survival (DSS) and Overall Survival (OS) in patients with HPV-negative head and neck squamous cell carcinoma (HNSCC). The study identifies the following as important predictors in both DSS and OS at diagnosis: high levels of IL-6, IL-31, IL-1RA, soluble IL-6Rα, and s-gp130. Further, it described their relationship with the TNM stage and history of smoking in predicting the outcome of the patients. This study truly demonstrates the unique prognostic ability of soluble cytokine receptors, as it suggests their prognostic uniqueness in the prediction of a clinical outcome. The study largely contributed to understanding the role of the acute-phase cascade in HNSCC and pointed out cytokine profiles as potential tools for therapy stratification and the development of new immunotherapies. The comprehensive analysis and clinical relevance of the findings offer a promising way for the development of new personalized, effective treatment approaches in HNSCC.

2.      General Comments

The exclusion criteria for HPV+ patients is somewhat incongruous. Since the authors suggest that different cytokine profiles are related to DSS and OS, wouldn’t HPV+ patients be the perfect control group since they have higher survival rates.

HNSCC tumors may be highly diverse, and it may be interesting to stratify the data based on tumor location. The authors even hint at this by stating in the introduction that HPV+ tumors have a higher survival rate if they originate in the oropharynx.

The inclusion of smoking history in the analysis is baffling. There is no significant association between smoking history and DSS or OS outcomes. This is not further elaborated upon in the discussion. Instead, smoking is mentioned as an inferior predictor of TMN stage (line 412f)?

3.      Specific Comments

The methods indicate that the study population is HPV- patients only, but the results from figure one show data from HPV+ patients. It may be advisable to provide HPV+ patient characteristics perhaps in a supplemental file. At least the number of HPV+ patients in figure one should be mentioned.

It might be necessary to include the actual concentration ranges of the measured cytokines and what the concentration cut-offs were for each cytokine when quartiles were constructed as in the results in figure 2. This is a major concern as it doesn’t allow to estimate if the concentration range itself is within normal parameters compared to a healthy or HPV+ cohort.

It may be beneficial to provide additional information on how smoking history was incorporated into the Cox Multivariate analysis, since Table 1 indicates five categories, or is it broken down by median pack-years?

The TANIS score method is not included in the methodology and should be described there.

Comments on the Quality of English Language

The verbiage is generally acceptable. It is more that some of the writing itself is unclear. 

For example, the authors often refer to their findings as having a significant impact on survival but fail to indicate if that impact increases or decreases survival. While I understand that all their results show a decrease in survival it is difficult to get that at first reading. 

Author Response

The exclusion criteria for HPV+ patients is somewhat incongruous. Since the authors suggest that different cytokine profiles are related to DSS and OS, wouldn’t HPV+ patients be the perfect control group since they have higher survival rates.

Thank you for the comment regarding the exclusion criteria for HPV(+) patients. We appreciate your insight into the potential benefits of including HPV(+) as a control group. As depicted in Figure 1. We have illustrated the difference between HPV(−) and HPV(+) in terms of DSS and OS. However, as stated in the first sentence of the Results section, the limited number of DSS/OS cases in HPV(+) patients (only 3 DSS cases and one non-DSS case) indeed posed a challenge for statistical analysis. It is noteworthy that none of the presently tested cytokines/soluble cytokine receptors, except IL-6 when binomially analyzed, predicted HPV(+) DSS/OS survival. To address this, we have added section 3.1.1. in the Results to clarify this limitation. We believe that including more survival curves from the tumor HPV(+) patients would not provide significant additional information. Therefore, we have chosen to present only results from tumor HPV(−) patients in our paper. To draw meaningful conclusions about HPV(+) patients,  extending the patient observation period or increasing  the patient number is crucial. We are currently in the process of doing so and plan to publish future results if they allow. We have included a paragraph addressing this in the Discussion section..

HNSCC tumors may be highly diverse, and it may be interesting to stratify the data based on tumor location. The authors even hint at this by stating in the introduction that HPV+ tumors have a higher survival rate if they originate in the oropharynx.

Thank you for the suggestion regarding stratifying the data based on tumor location. While we appreciate the potential value of this approach, we have decided not to implement it in the current study to due limitations in our sample size and all the data already presented. As indicated in Table 1, above 70% of the patients in our cohort are from oral cavity. However, we will keep this suggestion in mind for future research endeavors, particularly as we aim to study a larger cohort of patients.

The inclusion of smoking history in the analysis is baffling. There is no significant association between smoking history and DSS or OS outcomes. This is not further elaborated upon in the discussion. Instead, smoking is mentioned as an inferior predictor of TMN stage (line 412f)?

Thank you for bringing attention to the issue regarding the significance of smoking in relation to 5-year DSS or OS outcomes within our cohort. We understand the complexity of interpreting these findings.  In Table 1, we present the smoking data of the cohort, with over 60% identified as tobacco users. Analyzing the smoking data binomially indeed reveals a significant smoking-related survival prediction. Introducing this variable to the Cox Survival analysis instead of using the employed smoking did not alter the results regarding soluble cytokine survival predictions. Additionally, Kaplan-Meier log-rank comparison based on 5-point results shows a noticeable survival trend. We acknowledge, as pointed out, the lack of statistical significance for 5-year DSS. We state, however, in the Discussion that the cytokine survival predictions are largely secondary to a combination of TNM stage and smoking as shown in the Cox analyses. Notably, even after adjusting for these factors, the predictive value of soluble interleukin receptor for survival remains evident.

  1. Specific Comments

The methods indicate that the study population is HPV- patients only, but the results from figure one show data from HPV+ patients. It may be advisable to provide HPV+ patient characteristics perhaps in a supplemental file. At least the number of HPV+ patients in figure one should be mentioned.

Thank you for your comment regarding inclusion of data from HPV(+) patients in Figure 1 and the suggestion to provide HPV(+) patient characteristics in a supplementary file. The number of HPV(+) patients is already included in the Number at risk in Figure 1, but in our revised manuscript we have will ensure to mention the presence of HPV(+) in the text for clarity.  Additionally, we are pleased to inform you that we have included a supplementary table containing HPV(+) patient characteristics, as per your suggestion.

It might be necessary to include the actual concentration ranges of the measured cytokines and what the concentration cut-offs were for each cytokine when quartiles were constructed as in the results in figure 2. This is a major concern as it doesn’t allow to estimate if the concentration range itself is within normal parameters compared to a healthy or HPV+ cohort.

We appreciate your suggestion regarding inclusion of the actual concentration ranges, particularly in relation to quartile construction. In response, we have added Supplementary Table 2, which diplays the quartiles and minimum and maximum determined concentrations. For further information, we refer to our previously published paper addressing cytokine/soluble cytokine receptors concentrations from this cohort at diagnosis, dependent on HPV-status. We want to clarify that the findings of our study serve as a proof-of-concept, and the actual concentrations are specific to our assays. However, to address the need for additional context, we have included a ROC curve analysis with the actual concentrations from the IL-6 assay in Figure 5. This analysis provides a valuable demonstration for the performance of our assays in discriminating between different outcomes. We believe that he inclusion of Figure 5 enhances he interpretability and relevance of our findings.

It may be beneficial to provide additional information on how smoking history was incorporated into the Cox Multivariate analysis, since Table 1 indicates five categories, or is it broken down by median pack-years?

We have used the five-level smoking data in the Cox regression analyses. This has been more clearly stated in the table legend.

The TANIS score method is not included in the methodology and should be described there.

Thank you for bringing this to our attention. We apologize for the oversight regarding the exclusion of the TANIS score method in the methodology section. TANIS is an abbreviation of “T and N integer stage” which is constructed as the sum of T and N stage of the patient.   In the revised manuscript, we have added this information.

Comments on the Quality of English Language

The verbiage is generally acceptable. It is more that some of the writing itself is unclear. 

We appreciate your acknowledgement of the overall acceptability of the language used. We apologize for any ambiguity. We will strive to provide concise and precise descriptions to ensure our findings are communicated effectively to our readers and we have also planned for the use of MDPIs language editing services for the revised edition of our manuscript.

Reviewer 2 Report

Comments and Suggestions for Authors

The acute-phase cascade (APC) is a key driver of cancer outcomes, particularly in head and neck squamous cell carcinoma (HNSCC). Plasma levels of cytokines within the Interleukin-6 and IL-1 families were determined in 87 HPV-negative HNSCC patients at diagnosis. The study found that increased plasma levels of IL-6, IL-31, IL-1RA, soluble (s)-IL-6Rα, and s-gp130 at diagnosis were predictors of both OS and DSS in HPV-negative HNSCC patients. The cytokine DSS/OS predictions were associated with TNM stage and smoking history, while soluble receptors IL-6R, gp130, and IL33R more uniquely predicted DSS/OS. High plasma levels of IL-6, IL-31, and IL-1RA, as well as soluble receptors IL-6R, gp130, and IL33R, predicted clinical outcomes, indicating their potential for general therapy and immune therapy stratification. The author provides more details on the data from the experiment to suggest this study and adequately addresses my concerns in this manuscript. This manuscript content is suitable and related studies have not yet been published. This manuscript content is suitable for publication.

1.              Some errors in the manuscript content need to be corrected.

Author Response

Thank you for your feedback. We are committed to addressing any language issues and will ensure that the manuscript meets the highest standards of clarity and coherence. We have employed MDPIs language editing services for the revised version of the manuscript.   

Round 2

Reviewer 1 Report

Comments and Suggestions for Authors

The authors have sufficiently addressed my concerns and recommendations.